# Effect of the COVID-19 Emergency on Physical Function among School-Aged Children

**DOI:** 10.3390/ijerph18189620

**Published:** 2021-09-13

**Authors:** Tadashi Ito, Hideshi Sugiura, Yuji Ito, Koji Noritake, Nobuhiko Ochi

**Affiliations:** 1Three-Dimensional Motion Analysis Room, Aichi Prefectural Mikawa Aoitori Medical and Rehabilitation Center for Developmental Disabilities, Okazaki 444-0002, Japan; 2Department of Integrated Health Sciences, Graduate School of Medicine, Nagoya University, Nagoya 461-8673, Japan; hsugiura@met.nagoya-u.ac.jp; 3Department of Pediatrics, Aichi Prefectural Mikawa Aoitori Medical and Rehabilitation Center for Developmental Disabilities, Okazaki 444-0002, Japan; yuji.ito@med.nagoya-u.ac.jp (Y.I.); aoi2pochi@yahoo.co.jp (N.O.); 4Department of Orthopedic Surgery, Aichi Prefectural Mikawa Aoitori Medical and Rehabilitation Center for Developmental Disabilities, Okazaki 444-0002, Japan; noritake@mikawa-aoitori.jp

**Keywords:** physical health, COVID-19, pandemic response, restrictions, balance, single-leg standing time, body fat, Gait Deviation Index, Japan

## Abstract

In April 2020, the Japanese government declared a state of emergency due to the novel coronavirus disease (COVID-19). Schools were closed and a stay-at-home order was issued in April and May 2020. This before-and-after study aimed to measure the effects of these COVID-19-related restrictions on physical function among Japanese children. The study included children aged 6–7 years, enrolled before and after the emergency declaration. Their body fat percentage, single-leg standing time, Gait Deviation Index score, and history of falls were compared. There were 56 and 54 children in the before and after groups, respectively. Children in the after group had a higher body fat percentage (*p* = 0.037), shorter single-leg standing time (*p* = 0.003), and a larger number of falls per month (*p* < 0.001) than those in the before group. In the logistic regression analysis, children in the after group had a significantly shorter single-leg standing time (odds ratio (OR): 0.985, 95% confidence interval (CI): 0.972−0.997, *p* = 0.013), a greater number of falls per month (OR: 1.899, 95% CI: 1.123−3.210, *p* = 0.017), and a higher body fat percentage (OR: 1.111, 95% CI: 1.016−1.215, *p* = 0.020) than those in the before group. The COVID-19 emergency restrictions had a negative effect on children’s physical function, especially on balance.

## 1. Introduction

The World Health Organization declared the novel coronavirus disease (COVID-19) a pandemic in March 2020. A state of emergency was declared by the Japanese government in April and May 2020 and people were requested to maintain social distancing and to stay at home. During the state of emergency, children were not allowed to participate in sports and outdoor activities [1]. In addition, in many prefectures in Japan, schools were closed from 2 March to 25 May 2020. Although these emergency responses were necessary to prevent the spread of infection, school closures limited the children’s opportunities to engage in physical activities such as walking to and from school and physical education [2]. Children who are physically inactive may experience health problems and poor physical function later in life [3,4,5]. Therefore, it is important to assess whether children show signs of decline in physical function as a result of the COVID-19 emergency restrictions.

Few studies have investigated the changes in physical function in children since the start of the COVID-19 pandemic using direct measurements. Since December 2018, we have been conducting systematic physical checks of children for health-screening purposes. Although these data were collected as part of a service and were not originally intended for research, the availability of the data provided a unique opportunity to study the effects of the COVID-19 emergency on the physical function of children in Japan.

In this study, we aimed to investigate the impacts of the COVID-19 emergency declaration on physical function in children.

## 2. Materials and Methods

### 2.1. Study Design and Study Population

The study had a before-and-after design with two separate groups, and was conducted in a city in Japan with a population of about 387,000. A total of 557 COVID-19 cases were reported in the city in 2020. Study participants were recruited from 2 of 48 elementary schools in the city between December 2018 and December 2020. Physical function assessments were conducted among children attending these schools from December 2018 to December 2020. The assessments were suspended from 2 March to 25 May 2020 due to the COVID-19 emergency. The assessments consisted of a medical examination conducted by a pediatric orthopedic surgeon and a pediatric neurologist, who administered questionnaires to the children’s parents, and measured body fat percentage, single-leg standing time (SLST), grip strength, and gait analysis. In addition, participants were screened for intellectual disabilities using the Raven’s Coloured Progressive Matrices [6] and the Picture Vocabulary Test−Revised (Nihon Bunka Kagakusha Co., Ltd., Tokyo, Japan).

A total of 129 children, aged 6−7 years were eligible for this study. The legal guardians of all participants provided written informed consent, and the children provided verbal assent for participation.

The exclusion criteria were as follows: orthopedic, neurological, respiratory, cardiovascular, ophthalmologic, or auditory abnormalities that could affect the results of physical function tests; intellectual disabilities based on substandard scores on the Raven’s Coloured Progressive Matrices and Picture Vocabulary Test-Revised; previously diagnosed autism spectrum disorder or attention-deficit hyperactivity disorder; incomplete data. Of the 129 children, 19 were excluded due to the above reasons leaving 110 participants eligible for inclusion in the analysis. Participants were divided into two groups: the “before the emergency declaration” group comprising 56 children who were assessed between December 2018 and 2 March 2020; the “after the emergency declaration” group comprising 54 children who were assessed between 25 May and 31 December 2020.

This study was prepared following the Strengthening the Reporting of Observational Studies in Epidemiology (STROBE) guidelines.

### 2.2. Data Collection

#### 2.2.1. Questionnaire

The number of falls was determined by asking participants to recall the number of times they had fallen in the past month while walking at a normal pace. In addition, the guardians of all participants completed the Japanese version of the Strengths and Difficulties Questionnaire [7] and the Pediatric Quality of Life Inventory (PedsQL) 4.0 [8], as part of the mental health and quality of life component of the assessment. The Japanese version of the World Health Organization Health Behavior in School-Aged Children survey was used to assess whether the physical activity performed was at least 60 min of moderate-vigorous physical activity per day for more than five days per week [9]. The number of meals was assessed by using a 7-day diet history of three meals per day [10]. In order to investigate the sports costs, participants were asked to fill in the sports costs per month. Sleep time per day was assessed by sleep history.

#### 2.2.2. Body Fat Percentage

A multi-frequency bioelectrical impedance analyzer (BIA; MC-780; Tanita, Tokyo, Japan) was used to evaluate body fat percentage [9]. Measurements were performed in the standing position. The children stood with the thumb and palm of each hand holding hand electrodes and the soles of their feet maintaining contact with anterior and posterior foot electrodes. Their arms were extended in skin-to-skin contact to avoid a relaxed standing position. The evaluation was completed within 15 s. The BIA resistance was evaluated at three electrical frequencies: 5, 50, and 250 kHz [9]. Analysis of bioelectrical impedance is non-invasive, economic, and convenient; previous studies have used it to evaluate the body compositions of children [11,12]. This analysis was performed at least 2 h after meals, as recommended in the instruction manual.

#### 2.2.3. Single-Leg Standing Time

The SLST test evaluates how long a child can remain standing on a single leg with their eyes open to measure static balance function [9,13,14]. The test uses the mean SLST (in seconds) of both legs. This test has a good inter-rater reliability and test-retest reproducibility [13,14]. Children were instructed to lift one foot from the floor while avoiding bracing the lifted leg against the supporting leg. The evaluation ended when the lifted leg or toes touched the floor, when the lifted leg touched the supporting leg, or after 120 s of successfully maintaining balance.

#### 2.2.4. Grip Strength

Grip strength (in kg), defined as the mean strength of both hands, was evaluated using an adjustable handheld dynamometer (GRIP-D; Takei Ltd., Niigata, Japan). Grip strength was evaluated once in both left and right hands while participants were in the sitting position with one elbow extended, forearm and wrist in a neutral position, and shoulder abducted and neutrally rotated [9]. The reliability of grip strength assessment is excellent in the elbow-extension position [15]. Children were instructed to squeeze the handle of the dynamometer as hard as they could and sustain the grip for 5 s [9]. Prior to grip strength assessment, they were permitted to perform one test trial as a learning attempt. The dominant hand grip strength was evaluated first, followed by the non-dominant hand.

#### 2.2.5. Gait Analysis

A physical therapist with over 10 years of experience in clinical gait analysis conducted the three-dimensional gait analysis using a motion analysis system with eight cameras and a sampling frequency of 100 Hz (MX-T 20S; Vicon, Oxford, UK). The Plug-In-Gait model (December 2018 to March 2020) and Conventional Gait Model 2.3 (June 2020 to December 2020) were used to measure gait by placing markers on the lower body [16,17]. The children were recorded as they walked barefoot at a self-controlled speed on force plates of 8-AMTI OPT (Advanced Mechanical Technology, Inc., Watertown, MA, USA) during three trials [9].

The Gait Deviation Index (GDI) was used to assess the gait pattern, which was evaluated using a Vicon current pipeline [17]. The mean GDI score was calculated based on the results of the three gait trials to analyze both legs [9]. The GDI is an important index that represents the overall gait kinematic data using numerical values; it is determined based on a score derived from the kinematic data points of the gait analysis of the angles between the pelvis, hip, knee, and ankle in the sagittal plane; the angle between the pelvis and hip in the frontal plane; the angles between the pelvis, hip, and foot progression in the horizontal plane [17].

### 2.3. Statistical Analyses

A sample size power analysis was conducted using G*Power (Heinrich Heine University of Düsseldorf, Düsseldorf, Germany) to determine the optimal sample size for a statistical power of 0.95, a two-tailed alpha of 0.05, and a large effect size (d = 0.8) [18,19]. Based on these assumptions, the desired sample size was estimated to be 88 participants with two equal-sized groups.

The normality of the distribution for each variable was confirmed using the Shapiro–Wilk test. A chi-squared test was used to compare the difference in the proportion of each sex in each group. Participant data are expressed as means (standard deviations) or medians (range) and were compared using the independent *t*-test or Mann–Whitney *U* test, where appropriate. Effect sizes were calculated using *r* or Cramer’s V statistics. Effect sizes with *r* = 0.1 or −0.1 were considered small; *r* = 0.3 or −0.3 as moderate; *r* = 0.5 or −0.5 as large.

Logistic regression was used to assess the association between physical function, number of falls, and the COVID-19 emergency declaration. Two-sided *p*-values < 0.05 were considered statistically significant. Multivariable logistic regression was used to determine the odds ratios (ORs) of physical function and number of falls in relation to the COVID-19 emergency declaration. In this analysis, group (before or after) was the dependent variable. All data analyses were performed using SPSS version 24.0 (IBM Corp., Armonk, NY, USA).

## 3. Results

The demographic characteristics of participants are summarized by group in Table 1. Children in the after group were found to have experienced a larger number of falls in the past month (*p* < 0.001) than those in the before group. Moreover, children in the after group were more physically active (*p* = 0.025) than those in the before group. There were no significant differences in the results of body mass index, the Strengths and Difficulties Questionnaire, PedsQL 4.0, number of meals, sports costs, or sleep time between the two groups.

The physical function measures of participants are summarized by group in Table 2. Children in the after group had a higher body fat percentage (*p* = 0.037) and shorter SLST (*p* = 0.003) than those in the before group. There were no significant differences in the results of grip strength and GDI between the two groups.

The logistic regression results of physical function according to time period are summarized in Table 3. In the logistic regression analysis, children in the after group had a significantly shorter SLST, a significantly greater number of falls per month, and a higher body fat percentage than children in the before group.

## 4. Discussion

In this observational study, children assessed after the emergency declaration had a significantly lower SLST, significantly more falls, significantly more physical activity, and a significantly higher body fat percentage than children in the before group. These findings highlight the importance of performing physical activities, such as walking to and from the school and physical education, to maintain physical function, especially balance.

A study of children aged 5−13 years in the United States found that children engaged in less physical activity and more sedentary behavior during the early stages of the COVID-19 pandemic than before the pandemic, but many began to use remote streaming services to engage in physical activity to prevent a decline in their physical functions [20]. A study from Germany reported that digital media is playing an important role for sports activities during the COVID-19 pandemic [21]. Programmatic strategies using these methods are required to promote physical activity among school-aged children to reduce the risk of poor balance and an associated increase in the number of falls during periods when there are restrictions on movement outside the home due to the pandemic. However, our findings are not similar to those of previous studies [22,23,24,25] that demonstrated a significant decrease in physical activity in children after the emergency declaration compared to that in children before the emergency declaration. In the present study, the total time of physical activity increased among children in Japan, which is in line with the results of studies from Belgium, the Czech Republic, Germany, and Spain [26,27,28,29]. Thus, different behaviors of children from different countries might be due to factors such as differences in policy restrictions and the number of COVID-19 infections found in children [26]. Another study reported that the locations of physical activity also changed drastically, with more children performing physical activity at home, in the garage, and on sidewalks and roads [20]. Therefore, it was suggested that restrictions in the locations of physical activity or the low quality of the exercise would negatively impact balance function.

Furthermore, no differences were observed in the number of meals, sports costs, or sleep time between the before- and after-emergency declaration groups. Hence, these factors were not affected by the short period of the emergency declaration in this study, indicating that children with restrictions on movement outside the home did not necessarily exhibit greater changes in their number of meals, sports costs, or sleep time.

Pietrobelli et al. [30] reported that an unfavorable change in the eating habits of Italian children with obesity was observed during the lockdown due to the COVID-19 pandemic in Italy. The investigators expressed concern that these changes would persist, leading to an increased prevalence of obesity after the state of lockdown would be lifted. Thus, it is important to pay attention to eating habits during the lockdown. In Japan, the COVID-19 pandemic is ongoing. There are still restrictions on sports activities and leaving the home, and thus on children’s opportunities for physical activity. Thus, home- and neighborhood-based physical activity programs to promote children’s physical function are needed.

There are some limitations to this study. First, the study was observational in design with two different groups, rather than a cohort followed over time; therefore, there may be some residual confounding in the risk estimates. Second, the study was a single-center study and children were aged 6–7 years; hence, these findings may not apply to children of other ages or children living in other regions of Japan or other countries. Third, we did not ask in detail about changes in the children’s dietary habits. Fourth, the data on falls relied on self-reported questionnaire data so there is a possibility of recall bias.

In conclusion, to prevent a decrease in children’s physical function due to restrictions on activity imposed as a result of the COVID-19 pandemic, promoting physical activity and good eating habits is desirable. Measures should be taken to promote home- and neighborhood-based physical activities and exercise during children’s leisure time to prevent and reverse the adverse effects of COVID-19 emergency restrictions on children’s physical function.

## Figures and Tables

**Table 1 ijerph-18-09620-t001:** Demographic characteristics of participants, before and after the COVID-19 emergency declaration.

Variable	Children before the Emergency Declaration (*N* = 56)	Children after the Emergency Declaration (*N* = 54)	*p* ^1^	Effect Size ^2^ (*r* or Cramer’s V)
Age (years), median (range)	7 (6–7)	7 (6–7)	0.131	−0.1
Sex, *n* (%)			0.698	0.04
Female	28 (50.0)	29 (53.7)		
Male	28 (50.0)	25 (46.3)		
Height (cm), mean (SD)	118.7 (6.0)	119.3 (5.7)	0.560	0.1
Weight (kg), median (range)	20.5 (16.1–31.5)	21.6 (16.2–35.4)	0.130	−0.1
Body mass index, median (range)	14.5 (13.0–19.2)	15.3 (12.2–22.5)	0.094	−0.2
Number of falls per month, median (range)	0 (0−8)	0 (0−30)	<0.001	−0.4
Strengths and Difficulties Questionnaire (points), median (range)	9 (1–25)	10 (1–20)	0.947	−0.01
PedsQL 4.0 (points), median (range)	93.5 (64.1–100)	95.7 (63.0–100)	0.368	−0.1
Physical activity (hour), median (range)	3.0 (0–10.3)	4.0 (0–12)	0.025	−0.2
Number of meals (time), median (range)	21 (18–27)	21 (14−21)	0.399	−0.1
Sports costs (yen), median (range)	6000 (0–20,000)	5000 (0–20,000)	0.614	−0.1
Sleep time per day (hour), median (range)	9 (8−10)	9 (7−11)	0.544	−0.1

^1^ The *p*-value for the difference in the proportion by sex was calculated using the chi-squared test; the *p*-value for height was calculated using the independent *t*-test; other *p*-values were calculated using the Mann–Whitney *U* test. ^2^ Effect sizes with *r* = 0.1 or −0.1 were considered small; *r* = 0.3 or −0.3 as moderate; *r* = 0.5 or −0.5 as large. PedsQL 4.0, Pediatric Quality of Life Inventory 4.0.

**Table 2 ijerph-18-09620-t002:** Physical function in participants, before and after the COVID-19 emergency declaration.

Variable	Children before the Emergency Declaration (*N* = 56)	Children after the Emergency Declaration (*N* = 54)	*p* ^1^	Effect Size ^2^ (*r* or Cramer’s V)
Body fat percentage (%)	10.1 (3.3–23.3)	12.1 (3.0–34.1)	0.037	−0.2
Single-leg standing time (s)	60.3 (2.6−120)	33.1 (5.3−120)	0.003	−0.3
Grip strength (kg)	8.5 (5.8–15.9)	8.0 (5.4–15.5)	0.203	−0.1
Gait Deviation Index (points)	93.7 (7.0)	95.5 (7.9)	0.207	0.1

Data are presented as means (standard deviations), frequency (%), or medians (ranges). ^1^ The *p*-value of the Gait Deviation Index was calculated using the independent *t*-test; other *p*-values were calculated using the Mann–Whitney *U* test. ^2^ Effect sizes with *r* = 0.1 or −0.1 were considered small; *r* = 0.3 or −0.3 as moderate; *r* = 0.5 or −0.5 as large.

**Table 3 ijerph-18-09620-t003:** Relationship between physical function, number of falls, and the COVID-19 emergency declaration.

Variable	β	SE	Wald	Odds Ratio (95% CI)	*p*
Single-leg standing time	−0.016	0.006	6.16	0.985 (0.972−0.997)	0.013
Number of falls per month	0.641	0.268	5.732	1.899 (1.123−3.210)	0.017
Body fat percentage	0.105	0.045	5.379	1.111 (1.016−1.215)	0.020
Gait Deviation Index	0.043	0.029	2.138	1.044 (0.986−1.106)	0.144
Grip strength	−0.042	0.122	0.116	0.959 (0.754−1.219)	0.733

There were 56 children tested before, and 54 children tested after the COVID-19 emergency declaration. In this analysis, the time period was the dependent variable (before the emergency declaration = 0, after the emergency declaration = 1), and the number of falls per month was an adjustment variable. β, partial regression coefficient; CI, confidence interval; SE, standard error.

## Data Availability

All relevant data are presented within the manuscript. All data are available from the corresponding author on reasonable request.

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
