# Peer review of "Effect of the COVID-19 Emergency on Physical Function among School-Aged Children"

_ijerph, 2021, doi:10.3390/ijerph18189620_

Round 1

Reviewer 1 Report

Coronavirus disease (COVID-19), caused by SARS-CoV-2, has resulted in a devastating threat to human society in terms of health, economy, and lifestyle. Multiple studies indicate that the harm and suffering that the coronavirus can cause to an individual is determined by some factors such as age, sex, race, medical conditions and the lifestyle of the individual during the pandemic. In this study, authors separated a total of 129 Japanese children, aged 6 to 7 years, into two groups, called the before group and the after group. The body fat percentage, single leg standing time, grip strength and gait were analyzed between the before and after pandemic groups. The authors found the increased body fat percentage and decreased single leg standing time in the after group, compared to the before group. This study discussed the impact of COVID-19 and related physical inactivity on school-aged children in Japan.

Some issues need to be addressed.

1, In this study, the authors separated two groups using different individuals. Multiple factors between these two groups might have been different, such as diet exhibit, exercise, house income, etc before the covid pandemic. The authors need to provide more data to prove the increased body weight and decreased single leg standing time resulting from COVID-19, rather than individual exhibits.

2, Diet is a major factor to change body weight. In this study, body weight increased in the after group compared to the before group. Did the author consider that the increased body weight resulting from a change of diet?

3, The less of single leg standing time was observed in the after group compared to the before group. Did it mean the fewer physical activities happened in the after group compared to the before group? If so, whether more data can be provided by the authors.

Author Response

Reviewer #1

The authors would like to thank the reviewer for his/her constructive critique that has helped improve the manuscript. We have made every effort to address the issues raised and to respond to all comments. The revisions are indicated in red font in the revised manuscript. Please find below, a detailed point-by-point response to the comments.

Comment

Coronavirus disease (COVID-19), caused by SARS-CoV-2, has resulted in a devastating threat to human society in terms of health, economy, and lifestyle. Multiple studies indicate that the harm and suffering that the coronavirus can cause to an individual is determined by some factors such as age, sex, race, medical conditions and the lifestyle of the individual during the pandemic. In this study, authors separated a total of 129 Japanese children, aged 6 to 7 years, into two groups, called the before group and the after group. The body fat percentage, single leg standing time, grip strength and gait were analyzed between the before and after pandemic groups. The authors found the increased body fat percentage and decreased single leg standing time in the after group, compared to the before group. This study discussed the impact of COVID-19 and related physical inactivity on school-aged children in Japan.

Response: Thank you for reviewing our manuscript with close scrutiny. We wish to express our deep appreciation to the reviewer for the insightful comments.

Comment

In this study, the authors separated two groups using different individuals. Multiple factors between these two groups might have been different, such as diet exhibit, exercise, house income, etc before the covid pandemic. The authors need to provide more data to prove the increased body weight and decreased single leg standing time resulting from COVID-19, rather than individual exhibits.

Response: Thank you for your comment. It is crucial to justify the results in this study by providing more data. The Japanese version of the World Health Organization Health Behavior in School-aged Children was used to assess whether the physical activity performed was at least 60 min of moderate-vigorous physical activity per day for more than five days per week [9]. The number of meals was assessed by using a 7-day diet history as three meals a day [10]. In order to investigate the sports costs, participants were asked to fill in the sports costs per month. Sleep time per day was assessed by sleep history. As the results, children in the after group had a higher physical activity (P=0.025) than those in the before group. There were no significant differences in the results of number of meals, sports costs, and sleep time between the two groups. We have added this information in the Data collection (Lines 91-97), Results (Lines 167-171), Discussion (Lines 197-199), and Table 1.

Comment

Diet is a major factor to change body weight. In this study, body weight increased in the after group compared to the before group. Did the author consider that the increased body weight resulting from a change of diet?

Response: We would like to thank the reviewer for the constructive comment. As per the reviewer’s suggestion, we have added this information to the sentences discussing the results of the number of meals.

“Furthermore, no differences were observed in the number of meals, sports costs, and sleep time between the before and after emergency declaration groups. Hence, these factors were not affected by the short period of emergency declaration in this study, indicating that children with restrictions on movement outside the home did not necessarily exhibit greater changes in their number of meals, sports costs, and sleep time.” (Lines 222-226)

 However, we could not account for detailed change of diet habits; we have now also revised the limitation section to avoid any potential confusion.

"Third, we did not ask in detail about changes in the children’s dietary habits." (Lines 240-241).

Comment

The less of single leg standing time was observed in the after group compared to the before group. Did it mean the fewer physical activities happened in the after group compared to the before group? If so, whether more data can be provided by the authors. Response: Thank you for pointing this out. We also believe that it would be beneficial to provide more data regarding the physical activity. We have added this information to the "Materials and Methods." We have also added information in the Results and Discussion sections in the manuscript.

“Moreover, children in the after group were more physically activity (P=0.025) than those in the before group.” (Lines 167-168)

“Meanwhile, a study from Germany has reported that digital media is playing an important role for sports activities during the COVID-19 pandemic [21].” (Lines 206-207)

 “However, our findings were not similar to previous studies [22, 23, 24, 25] demonstrating a significant decrease of physical activity in children after the emergency declaration compared to children before the emergency declaration. In the present study the total time of the physical activity increased among children in Japan, which is in line with the results of studies from Belgium, Czech, Germany, and Spain [26, 27, 28, 29]. Thus, different behaviors of children from different countries might be due to factors such as difference in policy restrictions and the number of COVID-19 infections in children [26]. Meanwhile, another study reported that the locations of physical activity also changed drastically, with more children performing physical activity at home, in the garage, and on sidewalks and roads [20]. Therefore, it was suggested that restrictions in locations of physical activity or low quality of the exercise would be bad for balance function.” (Lines 211-221)

Reviewer 2 Report

The manuscript by Ito et al., studied important collateral effects of COVID-19 restriction on physical functions in a cohort of Japanese children. Although, authors very nicely used different parameters like body fat percentage, single leg standing time, Gait Deviation Index score, and history of falls and conclude that COVID-19 emergency restrictions had a negative effect on children’s physical function. But the conclusion stated is an overstatement since the study is not longitudinal. Body mass index for pre and post covid restriction is also not a reliable parameter since it was not in the same cohort of children. Also, there are numerous other parameters like changes in eating habits, sleeping habits apart from physical activity that could contribute to the negative impact of lockdown which was not included in this study.  However, authors acknowledge the limitation of this study, but the parameter studied fall short to clearly support the conclusion.

Author Response

Reviewer #2

We would like to thank the reviewer for providing insightful comments regarding our manuscript. We express our sincere gratitude for the time and effort the reviewer has spent reviewing our manuscript. The comments and suggestions have helped us immensely improve our work.

Comments

The manuscript by Ito et al., studied important collateral effects of COVID-19 restriction on physical functions in a cohort of Japanese children. Although, authors very nicely used different parameters like body fat percentage, single leg standing time, Gait Deviation Index score, and history of falls and conclude that COVID-19 emergency restrictions had a negative effect on children’s physical function. But the conclusion stated is an overstatement since the study is not longitudinal. Body mass index for pre and post covid restriction is also not a reliable parameter since it was not in the same cohort of children. Also, there are numerous other parameters like changes in eating habits, sleeping habits apart from physical activity that could contribute to the negative impact of lockdown which was not included in this study. However, authors acknowledge the limitation of this study, but the parameter studied fall short to clearly support the conclusion.

Response: Thank you for raising this issue. We think that the reviewer’s concern is valid. This study is a cross-sectional study and does not clearly support the conclusion. However, we expect that the results of physical function assessment in children not from the same cohort are also significant. The longer the duration of the longitudinal study, the more influence motor function such as balance and body mass index may have on the growth and development in children. Therefore, we believe that comparative data from before and after the COVID-19 pandemic by cross-sectional studies between groups that take into account the effects of growth and development are also important. As per the reviewer’s suggestion, we have added this information to the sentences describing the results of the other parameters (number of meals, sports costs, and sleep time) apart from physical activity.

As per your suggestion, in future, we plan to use longitudinal research data to analyze whether the poor balance function and increased body fat percentage is correlated with the physical activity restrictions in COVID-19 pandemic of the participants. Furthermore, if we obtain relevant results, we would submit our study to the International Journal of Environmental Research and Public Health.

Round 2

Reviewer 2 Report

Authors have nicely and clearly addressed all my concerns and comments. I have no more comments. I think the manuscript is sufficiently improved.